# Data-driven Design of Randomized Control Trials with Guaranteed Treatment Effects

Santiago Cortes-Gomez [1]   Naveen Raman [1]   Aarti Singh [1]   Bryan Wilder [1]

## Abstract

Randomized controlled trials (RCTs) generate guarantees for treatment effects. However, RCTs often spend unnecessary resources exploring sub-optimal treatments, which can reduce the power of treatment guarantees. To address this, we propose a two-stage RCT design. In the first stage, a data-driven screening procedure prunes low-impact treatments, while the second stage focuses on developing high-probability lower bounds for the best-performing treatment effect. Unlike existing adaptive RCT frameworks, our method is simple enough to be implemented in scenarios with limited adaptivity. We derive optimal designs for two-stage RCTs and demonstrate how such designs can be implemented through sample splitting. Empirically, we demonstrate that two-stage designs improve upon single-stage approaches, especially for scenarios where domain knowledge is available through a prior. Our work is thus, a simple yet effective design for RCTs, optimizing for the ability to certify with high probability the largest possible treatment effect for at least one of the arms studied.

## 1. Introduction

Randomized controlled trials (RCTs) are the gold standard for measuring treatment effects (Hariton and Locascio, 2018). In a traditional single-stage RCT, the experimenter fixes a set of treatments up front and randomizes the samples across them using predetermined assignment probabilities. While this design simplifies implementation and analysis, it often spends significant samples exploring suboptimal arms.

Policy decisions on scaling or adopting programs often depend on proving sufficient effectiveness. To that end, single

stage RCTs are often deployed alongside policy programs in order to quantify the policy's direct impact. Since underperforming arms are likely to be discontinued, precisely quantifying their effect sizes at the cost of reducing the statistical power for higher-quality treatments is not worthwhile. However, single-stage trials cannot allocate more samples to high-performing arms to improve the estimate precision.

Motivated by these limitations, a growing body of work has focused on adaptive trials, which periodically update the probability of assigning units to each arm in order to focus on more promising arms. While adaptive designs can enhance statistical performance, their implementation in practice can be challenging. First, treatment assignment probabilities must be updated dynamically, potentially after each sample, which imposes significant logistical demands on practitioners conducting the trial. Furthermore, high degrees of adaptivity may even be infeasible when outcomes are delayed. For instance, in clinical trials for chronic disease treatments, effect observations may occur only after months or even years, making it impossible to repeatedly change assignment probabilities based on realized outcomes.

In this work we question whether complex, adaptive designs are needed for improved statistical performance. Instead, we propose two-stage designs, which are deliberately simplified yet still allow for enhanced statistical guarantees. In the first stage, the experimenter uniformly explores arms and selects a subset of arms to retain for the second stage. In the second stage, the experimenter uniformly randomizes once more over the arms that are retained and computes the highest possible lower bound for one of the remaining treatments. The goal is to return as large a lower bound as possible, which requires both identifying arms with high rewards and concentrating enough samples on these arms to quantify (*certify*) its outcome accurately.

We make three main contributions. First, we design a novel two-stage algorithm with the goal of computing a high probability lower bound for a high performant treatment effect. We theoretically analyze this algorithm's performance and prove that it approximates the optimal two-stage design. Second, we extend our formulation to the scenario where

---

*Equal contribution   [1]Department of Machine Learning, Carnegie Mellon University. Correspondence to: Santiago Cortes-Gomez <scortesg@cs.cmu.edu>.

*Proceedings of the 42nd International Conference on Machine Learning*, Vancouver, Canada. PMLR 267, 2025. Copyright 2025 by the author(s).

the experimenter has a prior over the arm means (Bayesian setting). We present algorithms for this case and prove corresponding approximation guarantees. Third, we empirically demonstrate that our two-stage designs outperform single-stage designs for both synthetic and real-world datasets, and can outperform more complex adaptive designs even when experimenters have access to an informative prior. [1]

## 1.1. Related work

Naturally, our work is related to best-arm identification (Jamieson and Nowak, 2014; Auer, 2002b) and top-K arm identification (Bubeck et al., 2013). However, our approach differs from these settings in two key ways: first, our initial screening for good arms may not include all of the best ones for a given size, and it may not even contain the best arm. This is because our method is not designed for exhaustive exploration if it sacrifices certifying the final estimated effect. An interesting comparison can be made though with finding the $\epsilon$ best arms (Mason et al., 2020) setup. These formulations output all arms within an $\epsilon$ distance of the best one. However, once $\epsilon$ is fixed, the formulation remains exhaustive, as it aims to identify all good arms determined by the hyperparameter. In contrast, our method is not designed to optimize for exhaustive exploration in this way. Perhaps closest to our work is Katz-Samuels and Jamieson (2020) where they study the sample complexity inherent of $\epsilon$ best arms and of finding a fixed number of arms larger than a threshold. The second problem is somewhat related to our setting as they don't care about finding all good arms above the given threshold, but instead just a subset of such arms. However, they still fix an absolute hyperparameter, which is the number of good arms they aim to identify (in addition to a threshold defining the good arms). In contrast, our work adaptively chooses the number of arms to generate the largest certificate.

A current topic in adaptive trials literature is the study of the two usually desired, yet conflicting goals, for adaptive designs. One is to optimize for the cumulative regret, i.e., the benefit to the participants *during* the experiment. The other is to optimize for the information gained via the experiment for selection and deployment of the best arm *after* the experiment. The latter can be quantified via, e.g., the simple regret, of the precision with which treatment effects can be estimated. Previous work (Bubeck et al., 2011) shows that these goals are irreconcilable; designs with lower cumulative regret trade off nearly one-for-one in the worst case with information gain objectives. Our focus in this paper is on designs which identify a high-performing arm with the strongest statistical power possible for future deployment (i.e., our aim is not to minimize cumulative regret). (Athey et al., 2022; Li et al., 2010; Bastani et al., 2021; Kasy and Sautmann, 2021; Deshmukh et al., 2018; Chambaz et al., 2017; Simchi-Levi and Wang, 2023; Fan and Glynn, 2021).

Finally, identifying estimands with statistically valid properties after an adaptive trial, such as our *certification*, remains an active area of research. Due to the correlations introduced by adaptive procedures, performing inference on estimands obtained after the trial is concluded, without having a follow-up independent data collection for the identified arms, is not straightforward. Solutions to this problem have emerged from safe anytime inference (Waudby-Smith et al., 2021) or by imposing additional conditions, as in batch-only inference (Chen and Andrews, 2023). Our goal, instead, is to rely on the standard uniform allocation RCT, that is widely accepted by practitioners, in each stage to generate a certificate.

## 2. Problem Formulation

We introduce the problem of finding good certificates for randomized control trials (RCTs) and propose our own two-stage RCT.

### 2.1. Introducing Treatment Effects Certificates

Many real-world scenarios are more concerned in producing a high probability lower bound on the impact of treatments rather than guaranteeing optimality. For example, in policy settings, practitioners aim to give a guarantee on the performance of a policy (e.g. the policy has a certain positive treatment effect) in order to justify a course of action dictated by it. Formally, consider a set of $n$ arms, each corresponding to a treatment, with means $\boldsymbol{\mu} = (\mu_1, \mu_2, \ldots, \mu_n)$ and distributions $D_{\mu_i}$. Here, each $i$ corresponds to a treatment whose effect we estimate through an RCT. Our objective is to produce a high-probability lower bound $l$ for an arm $\mu_i$. We define a certificate as follows:

**Definition 2.1. Certificate** - Let $l$ be an estimand such that $l \leq \mu_i$ with probability $1 - \delta$ for some $i \in [n]$. Then $l$ is a certificate for $\mu_i$.

By maximizing $l$, we ensure a *certified* high effect for some arm $i$. Naturally, we can compute certificates in a single-stage RCT by uniformly allocating data to each arm. Nevertheless, as we mentioned in the introduction, this approach wastes samples on unpromising arms, thereby reducing the tightness of $l$. On the other extreme, fully adaptive trials are often infeasible to deploy as RCTs are typically run in-batches with fixed budgets across many individuals.

Therefore, we propose a deliberately simplified two-stage RCT that captures the benefits of adaptivity without introducing unnecessary complexity. In our approach, the first stage filters out suboptimal arms using a policy $\pi$, while the second stage computes the certificate $l$ by allocating more

---

[1]We include all code and datasets at `hidden`

interventions to the high-performing arms, resulting in a more precise estimate of the certificate $l$.

Formally, let $T$ be our total budget (which refers to the total number of samples), $s_1$ be our budget in the first stage, and $s_2$ be our budget in the second stage, with $s_1 + s_2 = T$. We view $s_1$, $s_2$, and $T$ as fixed parameters dictated by real-world logistical constraints. Let our data for the first stage be $X_1, X_2, \ldots, X_{s_1}$, where $X_i \in \mathbb{R}^n$ and $Y_1, Y_2, \ldots, Y_{s_2}$ the data for the second stage. For both the first and second stages, we uniformly explore all arms, e.g. in the first stage, we explore each arm $\lfloor \frac{s_1}{n} \rfloor$ times. We prune arms from the first stage through a policy function, $\pi(X_1, X_2, \ldots, X_{s_1}) \subseteq [n]$, which maps first-stage results to a set of surviving arms. Let $\mathbf{X} = \{X_1, X_2, \ldots, X_{s_1}\}$ and $\mathbf{Y} = \{Y_1, Y_2, \ldots, Y_{s_2}\}$. Our objective is then

$$f_{\boldsymbol{\mu}}(\pi) = \mathbb{E}_{\mathbf{X}, \mathbf{Y} \sim \boldsymbol{\mu}} \left[ \max_{i \in \pi(\mathbf{X})} l_i(\mathbf{Y}) \right] \tag{1}$$

Our proposed objective focuses on neither finding the single best arm nor any subset of the $k$-best arms, but rather on obtaining the best possible policy $\pi$ that leads to the highest certificate in expectation. Intuitively, $\pi(\mathbf{X})$ should discard less promising arms from the first stage while using minimal data. This approach frees up the budget for promising arms, which can improve the tightness of certificates.

## 2.2. Certificate estimation

We estimate certificates $l$ through concentration inequalities on the second-stage data. Suppose that we have a policy $\pi(\mathbf{X})$ which selects a set of $k$ arms. Let $\overline{Y}_i$ denote the empirical mean from the second stage for arm $i$, the reader should recall that such average is taken over $\lfloor s_2/k \rfloor$ samples. Then when $\mathbf{Y}$ is bounded, we can estimate $l$ through Hoeffding's inequality and the union bound by showing that $\overline{Y}_i$ concentrates near $\mu_i$:

$$P \left( \max_{i \in [n]} |\overline{Y}_i - \mu_i| < \sqrt{\frac{k}{2s_2} log(\frac{2n}{\delta})} \right) \geq 1 - \delta \tag{2}$$

Our certificate is therefore $l = \max_{i \in \pi(\mathbf{X})} \overline{Y}_i - \sqrt{\frac{k}{2s_2} log(\frac{2k}{\delta})}$. A boundedness assumption is natural in many real-world RCTs and applies to RCTs where rewards are binary.

Analyzing the certificate, we find that a two-stage approach provides an advantage over a single-stage approach if:

$$\sqrt{\frac{1}{2|s_2/k|} \log \left( \frac{2k}{\delta} \right)} < \sqrt{\frac{1}{2|T/n|} \log \left( \frac{2k}{\delta} \right)}$$

We note that there are two competing terms when trying to maximize $l$ in Equation 2. Intuitively, a certificate is

maximized when a large set of arms is pruned, as this minimizes $k$, while still leaving sufficient budget to explore more promising arms. Simplifying, this means that $\lfloor s_2/k \rfloor > \lfloor T/n \rfloor$; such a result occurs because single-stage methods can use all $T/n$ samples per arm due to its lack of adaptivity, while we only use $s_2/k$. Therefore, a policy $\pi$ is more effective when a small subset of the total arms can be identified with a relatively small amount of data.

## 3. Finding optimal policies for Two-Stage RCTs

Developing two-stage designs requires finding a policy $\pi$ that maximizes Equation 1. We focus our analysis on the so-called top-$k$ policies because they are well known to have good performance in a wide variety of scenarios, and because top-$k$ policies are easy to explain to decision makers, making them ideal for practical implementation. In the following sections, first we propose a top-$k$ algorithm with approximation guarantees for the optimal policy within the top-$k$ class. Second, we show the conditions under which this policy class always contains an optimal policy.

### 3.1. Designing Top-K Policies

We begin by defining top-k policies.

**Definition 3.1. Top-K Policy** - Let $\sigma(\mathbf{X})$ be the descending ordering of arms by empirical mean from the first stage; that is $\bar{X}_{\sigma(\mathbf{X})_1} \geq \bar{X}_{\sigma(\mathbf{X})_2} \geq \ldots \geq \bar{X}_{\sigma(\mathbf{X})_n}$, where ties are broken randomly. A top-k policy outputs sets of the form $\pi(\mathbf{X}) = \{\sigma(\mathbf{X})_1, \sigma(\mathbf{X})_2 \ldots \sigma(\mathbf{X})_{k(\mathbf{X})}\}$ for some function $k(\mathbf{X})$.

Top-k policies are a natural approach to pruning arms, aiming to keep the $k$ best observed ones. However, the optimal $k$, which we denote $k^*$, is unknown. We propose a sample splitting design to estimate $k^*$.

We construct our sample splitting design by splitting the data from the first stage into two halves: training ($U$) and validation ($V$).

Our training half is used to compute empirical means and sort arms, while our validation half is used to compute certificate values for each value of $k$; essentially, we use the validation set $V$ to simulate the second stage.

We can then estimate the certificate value for different choices of $k$ and select the $k$ which maximizes certificates. We provide pseudocode below in Algorithm 1. To guarantee the performance of our sample splitting design, we compare our design against an optimal two-stage design that selects $k^*$ arms.

**Proposition 3.2.** *Let $\Delta_{ij} = \mu_{\sigma_i} - \mu_{\sigma_j}$. Let $\boldsymbol{\mu}$ be such that assumption 3.4 holds. Let $\sigma$ be the permutation of the indexes obtained from sorting the empirical means obtained in*

**Algorithm 1** Sample splitting design

1: **Input:** $s_1$ iid samples.
2: **Output:** Set $\pi(\mathbf{X})$
3: Split first stage data randomly into two sets: $U = \{x_1, ..., x_{\frac{s_1}{2}}\}$ and $V = \{z_1, ..., z_{\frac{s_1}{2}}\}$.
4: Compute $\bar{U}$, which is the average per-arm using data from $U$
5: Let $\sigma$ be the ordering of arms according to $\bar{U}$
6: Compute $\bar{V}$, which is the average per-arm using data from $V$
7: **for** $i = 1$ to k **do**
8: $\quad$ set $l_i = \text{argmax}_{j \in [i]} \bar{V}_{\sigma_j} - \sqrt{\frac{i}{2s_2} log(\frac{2n}{\delta})}$
9: **end for**
10: $k = \text{argmax}_{i \in [n]} l_i$
11: Let $\sigma'(\mathbf{X})$ be ordering of arms according $U \cup V$
12: **return** $\pi(\mathbf{X}) = \{\sigma'(\mathbf{X})_1, \sigma'(\mathbf{X})_2, \ldots, \sigma'(\mathbf{X})_k\}$

the first stage in descending order. Let $c(i) = \sqrt{2log(\frac{2i}{\delta})\frac{i}{s_2}}$. Then, conditioned on $\mathbf{X}$ for any top-k policy $\pi^k$ obtain as an output of algorithm 1, the expected value of its certificate $f(\pi^k)$ is bounded below by

$$
f(\pi^*) - \sum_{i=1}^{k^*} \exp\left(\frac{-\left[\Delta_{k^*i} - (c(k^*) - c(i))\right]^2 s_1}{n}\right) \\
\times [\Delta_{k^*i} - (c(k^*) - c(i))] \\
- \sum_{i=k^*+1}^{n} \exp\left(\frac{-\Delta_{ik^*}^2 s_1}{n}\right) [\Delta_{k^*i} - (c(k^*) - c(i))]
$$

(3)

This result is analogous to guarantees on simple regret (Bubeck et al., 2011), except gaps between means are replaced with gaps between lower bounds. Notably, although the bound depends on $k^*$ our design (and thus the guarantee) does not, making our procedure adaptive. In particular, if $k^* = n$, the bound will be identical to the one obtained from a single-stage approach. Notably, as a consequence of the gaps being part of the bound, the tension between $s_1$ and $s_2$ becomes explicit; higher $s_1$ leads to exponential improvements on the bounds at the cost of the loosening it by $\frac{1}{\sqrt{s_2}}$.

### 3.2. Optimality of top-k policies

It remains an open question in this paper how to compute an optimal policy for our two-stage design (whether or not such a policy is top-$k$). However, despite the combinatorial complexity of finding the optimal policy, it turns out that a simple yet sufficient condition guarantees the global optimality of top-$k$ policies.

**Definition 3.3. First-order stochastic dominance** A random variable $A$ is said to first order stochastically dominate

a random variable $B$ if $P(B \geq x) \leq P(A \geq x)$ for all $x$.

**Assumption 3.4.** For any $i, j$ such that $\mu_i \geq \mu_j$, $D_{\mu_i}$ first-order stochastically dominates $D_{\mu_j}$

We aim to show that when outcome distributions respect stochastic domination, the optimal policy for Equation 1 can be written as a top-k policy. To do so, we first demonstrate that stochastic domination in the true means $\mu_i$ corresponds to stochastic domination in the empirical means $\bar{X}_i$.

**Lemma 3.5.** *Let $\sigma$ be the descending ordering of arms by empirical mean observed in the first stage. Then, for any $i < j$, $D_{\mu_{\sigma_i(\mathbf{X})}}$ first-order stochastically dominates, $D_{\mu_{\sigma_i(\mathbf{X})}}$*

This allows us to translate any policy into a top-k counterpart which achieves higher value for $f(\pi)$. We prove this through properties of stochastic domination and formally prove all Theorems in Appendix F. We next show how this allows us to convert any policy into a top-k variant of the policy.

**Lemma 3.6.** *Fix an arbitrary policy $\pi$. Define it's top-k counterpart $\pi'$ to be the top-k policy with $k(\mathbf{X}) = |\pi(\mathbf{X})|$. That is, it selects the same number of arms as $\pi$ does for every realization of the trial, but it selects those arms to be the ones with the largest empirical mean. Then $f_{\boldsymbol{\mu}}(\pi') \geq f_{\boldsymbol{\mu}}(\pi)$.*

The main idea is as follows: consider any set of arms $\pi(\mathbf{X})$ selected by the original policy. If there is an arm outside of $\pi(\mathbf{X})$ with strictly higher empirical mean than contained within $\pi(\mathbf{X})$, we can only do better by swapping in the arm with higher empirical mean. Formally, this follows because stochastic dominance implies a corresponding ordering on any monotone function of a random variable, including the max function which appears in our objective function. Together, these results imply the existence of an optimal top-k policy:

**Theorem 3.7.** *Let $\pi^*$ the policy that maximizes $f_{\boldsymbol{\mu}}$. There exists a top-k policy $\pi$ such that $f_{\boldsymbol{\mu}}(\pi^*) = f_{\boldsymbol{\mu}}(\pi)$.*

It is worth noticing that this result applies to a wide variety of setups. For example, it holds when all treatment arms have binary outcomes or when outcomes are normally distributed with the same variance, both of which are commonly encountered in practice. Additionally, we present satisfactory experimental results even for cases besides the two previous mentioned ones, for example, where the arms are normally distributed with different variances.

### 3.3. Incorporating Priors

So far, we have developed algorithms that do not rely on any prior information about $\boldsymbol{\mu}$ and aim to approximate the optimal policy solely using information gathered during the trial. In this section, we modify this setting slightly by allowing for prior information on the treatment effects.

Such a prior not only enables the creation of algorithms that leverage Bayesian reasoning, but also removes the need for the assumption of stochastic domination that was previously required.

In many practical settings, the experimenter may have an informative prior over $\boldsymbol{\mu}$ given previous work in the domain. For example, many meta-analyses have quantified the distribution of reported effect sizes for interventions in domains such as education (Evans and Yuan, 2022), medicine (Greising et al., 2009), and development (Iwasaki and Tokunaga, 2014). Although effect sizes for novel treatments are unknown, the experimenter may be able to improve their design by modeling them as drawn from such a prior distribution.

Formally, we assume access to a joint prior, $\mathcal{P}$, so that the mean of each arm is distributed according to $\mu_1, \mu_2, \ldots, \mu_n \sim \mathcal{P}$. Our goal then, is to find a policy that maximizes the certificate when arms are distributed according to the prior, that is finding the policy $\pi^*$ that maximizes:

$$f(\pi) = \mathbb{E}_{\boldsymbol{\mu} \sim \mathcal{P}} \left[ \mathbb{E}_{\mathbf{X}, \mathbf{Y} \sim \boldsymbol{\mu}} \left[ \max_{i \in \pi(\mathbf{X})} l_i(\mathbf{Y}) \right] \right] \quad (4)$$

To find the optimal policy, we note that $\pi^*$ can be computed for each $\mathbf{X}$ by optimizing over the posterior $\mathcal{P}(\boldsymbol{\mu}|\mathbf{X})$, in other words by maximizing,

$$f_{\mathbf{X}}(\pi) = \mathbb{E}_{\boldsymbol{\mu} \sim \mathcal{P}(\boldsymbol{\mu}|\mathbf{X})} \left[ \mathbb{E}_{\mathbf{Y} \sim \boldsymbol{\mu}} \left[ \max_{i \in \pi(\mathbf{X})} l_i(\mathbf{Y})|\mathbf{X} \right] \right] \quad (5)$$

Importantly, this certificate $l$ still has the same frequentist coverage guarantees; the Bayesian prior is used only to improve the power of the design.

We develop a prior-based design to greedily compute $\pi(\mathbf{X})$, assuming sampling access to the posterior distribution. The sampling can be implemented through many different Bayesian inference approaches; in this paper, we ignore the specifics, and let this be a black box. The procedure begins by sampling $d$ values of $\boldsymbol{\mu}$ from the posterior. Then, for every set $B^1 = \{b\} \subset [n]$ of size 1, using the sample-average certificate obtained from the posterior draw, we estimate the value of $\mathbb{E}_{\mathbf{Y} \sim \boldsymbol{\mu}} \left[ \max_{i \in B^1} l_i(\mathbf{Y}) \right]$. This process is repeated for all possible sets of size two, $B^2 = \{b, i\}$ for $i \in [n] \setminus \{b\}$, to greedily construct $B^2$, a candidate set of size two. The design proceeds iteratively, adding elements to $B^i$ until $|B^i| = n$. We select the set, $B^k$, with the largest estimated certificate, where such an estimate is done through resampling from the posterior.

Through reduction to submodular optimization, we demonstrate approximation guarantees for our design. Formally,

**Theorem 3.8.** *Let $\hat{\pi}$ be the policy obtained by our proposed design using $d$ samples from the posterior $\mathcal{P}(\boldsymbol{\mu}|\mathbf{X})$. Then* $f(\hat{\pi}) \geq f(\pi^*)(1 - 1/e - \epsilon)$, *where* $\epsilon = O\left(\sqrt{\log\left(\frac{1}{\delta}\right) d}\right)$.

The idea behind the proof is to leverage the property that greedy algorithms are $1 - 1/e$ optimal for monotonic submodular optimization, and we demonstrate that our situation does in fact match this type of problem. Furthermore, we present a hardness result, showing that the $1 - 1/e$ bound is tight; details on this result can be found in Appendix F.

## 4. Experiments

We assess our two-stage RCT design with both synthetic and real-world datasets.

**Synthetic Dataset and Setup**   We construct a synthetic dataset to evaluate our two-stage RCT designs. We sample arm means, $\boldsymbol{\mu}$, from a uniform 0-1 distribution (we experiment with other choices in Appendix E). Arms have Bernoulli outcomes with mean $\mu_i$, which simulates settings where treatment are successful with probability $\mu_i$. We fix $n = 10$ (we find similar results for other $n$ in Appendix D) and $\delta = 0.1$ (and find similar results for other $\delta$ in Appendix B). We compare the following RCT designs

1. **Random** - Two-stage top-k design + random $k$

2. **Best Arm** - Two-stage design with $k = 1$

3. **Single-stage** - A single-stage design which uniformly randomizes the entire budget $T$ over the arms.

4. **Sample Split** - Our proposed two-stage method uses the first stage to prune arms and the second stage to compute certificates

5. **Omniscient** - A two-stage method which computes $k^*$ with knowledge of $\mu$. In particular, $k^* = \text{argmax}_{i \in \pi(\mathbf{X})} \mu_i - \sqrt{log(\frac{1}{\delta})\frac{i}{2s_2}}$. Such a design serves as an upper bound on the performance of any two-stage design.

We compare designs by measuring normalized certificates, which is the ratio of $l$ to $\max_i \mu_i$. We average results over 15 seeds and 100 runs per seed; seeds sample values of $\boldsymbol{\mu}$, while runs sample values for $\mathbf{X}$ and $\mathbf{Y}$.

**Comparison against Single-Stage Designs**   We compare our two-stage design against baselines and find that our sample splitting methods improve upon baselines. In Figure 1, we find that our sample splitting methods outperform single-stage methods across first-stage percentages. When $s_1$ is 30% of the budget, sample splitting methods outperform

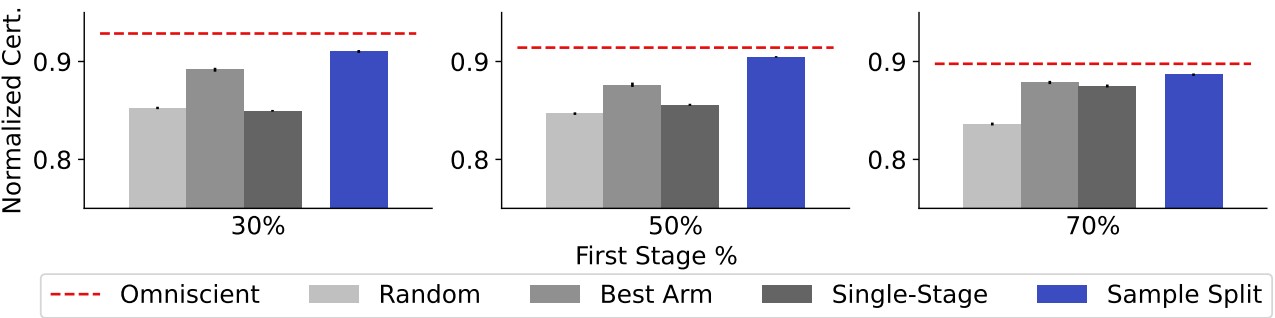

*Figure 1.* Our sample splitting design outperforms all baselines across first-stage sizes. The largest improvement occurs when the first stage is small, as this leaves a budget for the second stage to compute certificates.

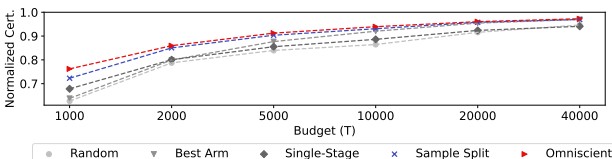

*Figure 2.* Sample splitting designs perform well for all values of $T$. When $T$ is large, the sample splitting approaches the optimal two-stage policy (omniscient).

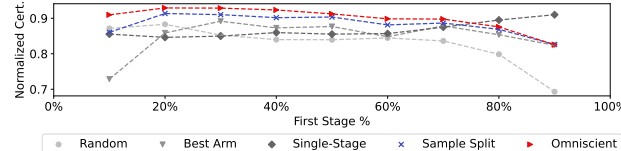

*Figure 3.* Single-stage methods perform best when between 20% and 70% of the budget is spent in the first stage, as this allows for arms to be pruned, and a certificate to be generated in the second stage.

single-stage methods by 7%, while when $s_1$ is 70% of the budget, sample splitting outperform single-stage methods by only 1%. With $s_1$ is large, $s_2$ is smaller, and so sampling splitting designs have less data to use for calculating certificates. However, when $s_1$ and $s_2$ are balanced, we can prune many arms in the first stage, while leaving sufficient time to find certificate values.

**Impact of Budget**   To understand our designs across choices of $s_1$ and $T$, we compare performance across experimental designs, both when a) letting $s_1 = s_2$, while varying $T$, and b) fixing $T$ while varying $s_1$.

In Figure 2, we find that sample splitting designs are significantly better than all baselines when $T \leq 10000$ ($p < 10^{-11}$). While best arm designs are 3% better than sample splitting designs for $T = 40000$, best arm designs are 13% worse than sample splitting for $T = 1000$. When comparing against the omniscient certificate, we see that sample splitting designs approach the omniscient policy for large budgets, as they are within 0.5% for $T = 40000$.

We compare our two-stage methods against baselines when varying the ratio of $s_1$ to $T$. In Figure 3, we show that sample splitting excels with large first-stage sizes, whereas single-stage methods perform best when $\frac{s_1}{T} \geq 70\%$. When the first stage is much larger than the second stage, two-

stage methods use only data from the second stage, while single-stage methods use data from both stages. This occurs due to the adaptivity from the second stage, which prevents us from applying Hoeffdings bound to data from both stages due to the lack of the iid property.

**Comparison with Adaptive Designs**   We compare our approach against adaptive designs which can potentially improve the guarantees at the cost of complexity. We detail the adaptive methods below:

1. **Two-Stage Successive Elimination** - We perform successive elimination in-batch, by first doing uniform exploration in the first stage, then running successive elimination to prune arms (Even-Dar et al., 2006). We then re-run uniform exploration in the second stage.

2. **Two-Stage Thompson Sampling** - In the first stage, two-stage Thompson Sampling performs uniform exploration, and in the second stage, it performs non-uniform exploration in the second round based on probabilities from Thompson sampling probabilities with a uniform prior (Russo et al., 2018).

3. **Successive Elimination** - We run the successive elimination algorithm, with a budget of $T$.

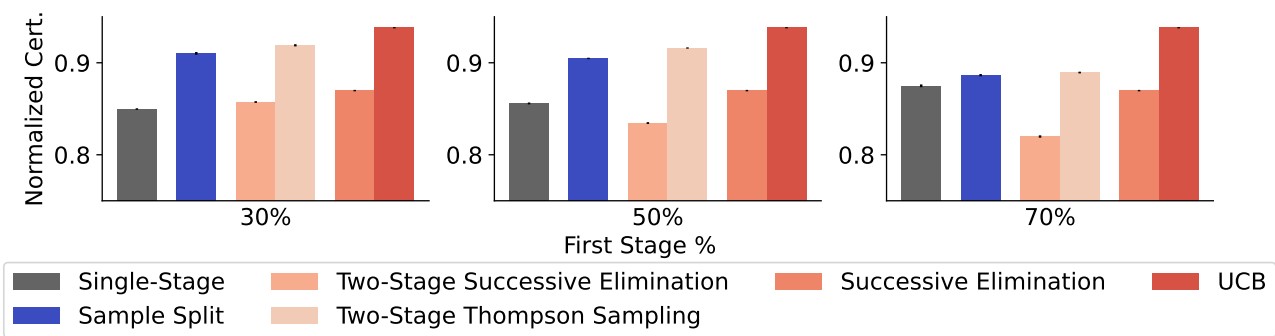

*Figure 4.* Sample splitting designs can close the gap between single-stage and adaptive designs (such as UCB), and serve as a middle ground in performance and complexity. This is best seen when the first stage is 30%, as sample split methods can capture up to 69% of the improvement between single-stage and UCB designs.

4. **UCB** - We run the upper confidence bound (UCB) algorithm with budget $T$ (Auer, 2002a). Note that UCB has a much higher degree of adaptivity: it updates assignment probabilities $T-1$ times, compared to once for the two-stage designs.

We compare our design to adaptive approaches in Figure 4. We find that non-uniform allocation probabilities have limited benefits. Sample splitting performs within 1.5% of two-stage Thompson Sampling. Sample splitting also performs significantly better than both the two-stage and fully adaptive versions of Successive Elimination. UCB is the strongest fully adaptive design and can perform better than less-adaptive designs. However, the best sample splitting design (where $s_1$ is 30% of the budget) comes relatively close: Sample Splitting performs within 3% of UCB, capturing up to 69% of the improvement between the single-stage design and UCB. Practioners can capture much of the value of the most complex, highly adaptive design by using a properly configured two-stage, framework.

**Bayesian Setting** We explore whether knowledge of a prior distribution can improve the certificates discovered by two-stage designs. We compare our prior-based design against both sample splitting and baseline design on a synthetic dataset. To construct such a dataset, we let $\boldsymbol{\mu}$ be distributed according to a $\beta$ distribution, fixing $\alpha = 1$ and varying $\beta \in \{1, 2, 4\}$. Higher $\beta$ indicates a more informative prior.

Figure 5 shows that informative priors (large $\beta$ allow prior-based methods to perform well, as they slightly exceed the performance of UCB at $\beta = 2$ and improve upon UCB by 18% at $\beta = 4$. When available, informative priors of effect sizes contribute more than the ability to incorporate high degrees of adaptivity.

We compare the performance of our designs under prior misspecification by adding Gaussian noise to $\boldsymbol{\mu}$ with $\alpha = \beta =$

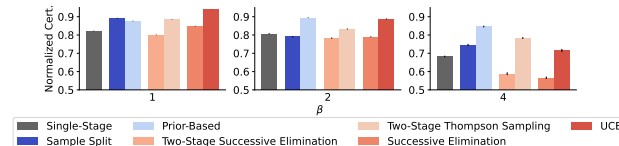

*Figure 5.* When priors are informative, correlating to large $\beta$, prior-based methods can improve upon all designs, including adaptive designs such as UCB and two-stage Thompson Sampling.

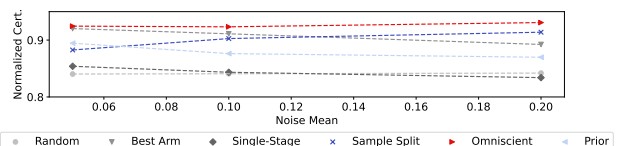

*Figure 6.* Prior-based designs are sensitive to the mean of the noise; directional noise (corresponding to larger values for noise) leads to degrading performance, especially compared with sample splitting designs.

1. We fix the noise variance to be $0.01$ and vary the mean in $\{0.05, 0.1, 0.2\}$. In Figure 6, we see that when the prior is minimally misspecified, prior-based designs outperform sample split. However, prior-based designs fare poorly with increasing misspecification, demonstrating that prior-based are not robust to large degrees of misspecification.

**Real-World Experiments** We run semi-synthetic experiments where effect sizes are drawn accordingly to a real-world distribution drawn from a meta-analysis of treatments in gerontology (Greising et al., 2009). We retrieved 75 effect sizes from the meta-analysis and set the prior on $\boldsymbol{\mu}$ to be uniform over these values. Since effect sizes are reported as Cohen's d (a standardized metric), we model the outcome distribution $D_{\mu_i}$ as a normal with mean $\mu_i$ and standard

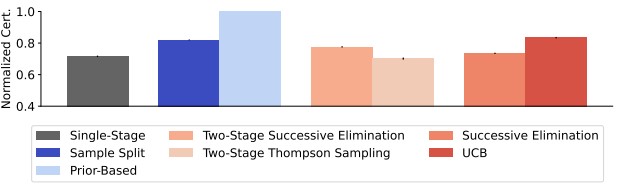

*Figure 7.* On a real-world genertology dataset, we see that using prior-based methods improves the certificates generated by RCTs, even compared to adaptive methods. This reflects the added benefit we get when using domain knowledge for real-world RCTs.

deviation 1. For the certificate $\ell$, we use the corresponding tail bound for Sub-Gaussian variables:

$$P\left(|\bar{Y_i} - \mu_i| \leq \sqrt{\frac{2k}{s_2}\log(\frac{2k}{\delta})}\right) \geq 1 - \delta \qquad (6)$$

In Figure 7, we see that prior-based methods perform best, beating even adaptive methods like UCB by 23%. Restricting our attention to prior-free designs, sample splitting performs within 2% of UCB. This verifies that the main conclusions from the synthetic experiments continue to hold on a real-world distribution: two-stage designs based on our sample splitting procedure can nearly match the performance of fully adaptive designs, and with access to the prior, Bayesian two-stage designs perform significantly better.

## 5. Conclusion and Limitations

Traditional single-stage RCTs spend unnecessary resources exploring sub-optimal arms, while fully adaptive procedures are often costly. To improve on these designs, we study two-stage RCTs, which improves guarantees from single-stage RCTs, while deliberately maintaining simplicity. We develop a top-k policy for designing such RCTs and demonstrate the optimality of top-k two-stage designs under a stochastic dominance ordering. We empirically demonstrate that our two-stage RCT can significantly improve guarantees compared with single-stage RCTs, and can even outperform adaptive methods in Bayesian settings. By using two-stage RCTs, real-world studies can improve guarantees without increasing complexity. Better understanding the level of adaptivity available can also instruct us how multi-stage RCTs can be better designed. For example, with a larger adaptivity budget can allow us to construct three or four-stage RCTs, which could potentially improve the guarantees delivered by our algorithms. Our work demonstrates the added benefits when introducing two-stage RCTs, and how these methods can be customized depending on the situation.

## Impact Statement

Our work is highly robust to deployment, as it is designed to certify, in the most data-efficient manner possible, a worst-case scenario guarantee. However, the conclusions drawn from the results, as well as the interpretive reasoning (hermeneutics) behind a measured effect, will always depend on practitioners with appropriate domain knowledge.

## Acknowledgments

Santiago Cortes-Gomez was supported by the AISDM Institute. Naveen Raman was supported by the National Science Foundation Graduate Research Fellowship (NSF GRFP). We would like to thank Fei Fang for valuable discussions during the development of this work.

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

## A. Certificate Spread

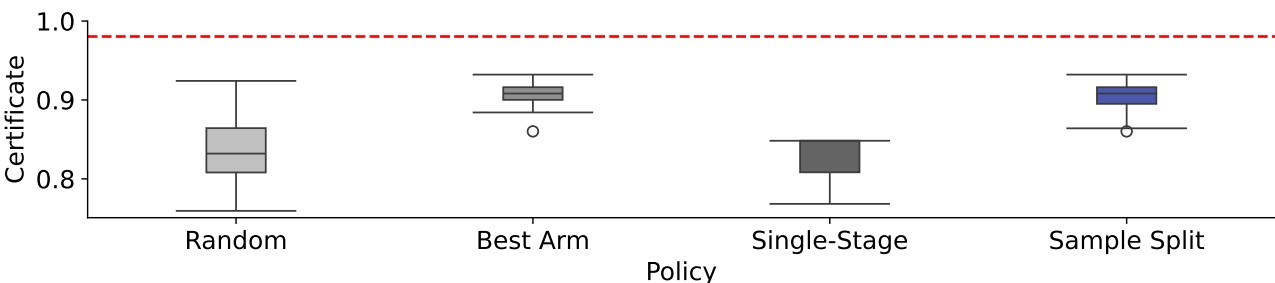

*Figure 8.* We compare the distribution of normalized certificates across various designs. Sample splitting designs have a lower spread and higher average certificate compared with single-stage designs, showing that sample splitting improves upon single-stage designs.

To understand how the performance of our designs varies across random seeds, we plot the distribution of normalized certificates in Figure 8 with $s_1 = s_2 = 500$. We find that our sample splitting design has a lower spread compared to a single-stage design. In this scenario, even the 25th percentile for the sample splitting design does better than the 75th percentile for the single-stage design, showing the advantage of using a sample splitting design.

## B. Impact of Model Confidence

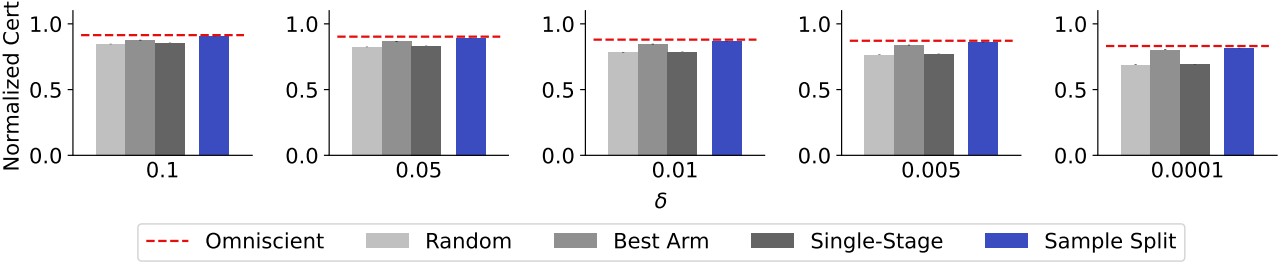

*Figure 9.* We compare the performance across designs when varying $\delta$. We find that for all values of $\delta$, the sample splitting algorithm performs best, with the gap between sample splitting and best arm becoming smaller for smaller $\delta$.

The choice of $\delta$ impacts the probability with which the certificate serves as a lower bound. Lower $\delta$ corresponds to scenarios where higher probability certificates are desired, which naturally leads to looser lower bounds. We compare the performance of our designs across values of $\delta$ in Figure 9, and find that sample splitting policies perform well across choices of $\delta$. As $\delta$ becomes smaller, which corresponds to looser lower bounds, we find that the best arm policy approaches the performance of sample splitting policies. However, the opposite scenario is seen with single-stage policies, which perform worse for smaller $\delta$. Such a trend occurs because smaller $\delta$ makes it more important to discard low-probability arms, as this can allow for more budget to generate certificates.

## C. Additional Stages

We evaluate the impact of extending two-stage designs to larger-stage designs. We compare designs in two ways; the first is using our sample splitting algorithm, while only using the last stage to compute certificates, while the second is using our sample splitting algorithm while using all the stages to compute certificates (which we call "sample split total"). While only the last stage preserves the i.i.d property needed for concentration inequalities such as Hoeffdings, real-world applications frequently use all the data collected to generate certificates.

In Figure 10, we find that using more stages is only helpful when using all the data to generate certificates. In the setting when generating data from only the last stage, we find that two-stage designs are optimal. Moreover, while more stages can

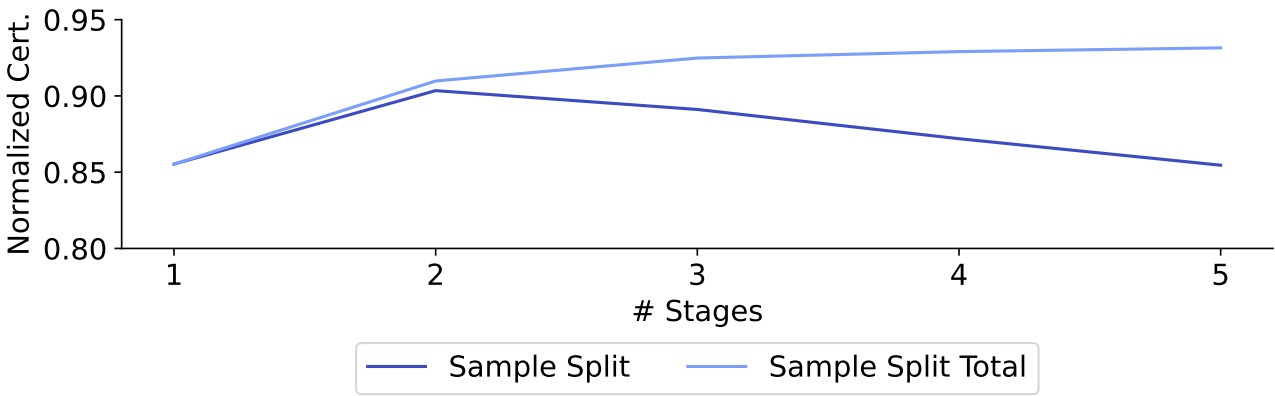

*Figure 10.* We extend two-stage methods up to five stages and compare the performance when using either (a) sample splitting methods, generating certificates with only the last stage, or (b) sample splitting methods using all of the data. When using only the last stage data, we find that two-stage designs are optimal, while when using all of the data, we find that using more stages increases performance.

improve certificates when using all stages to generate certificates, the impact of more stages diminishes quickly beyond two stages.

## D. Varying the number of arms

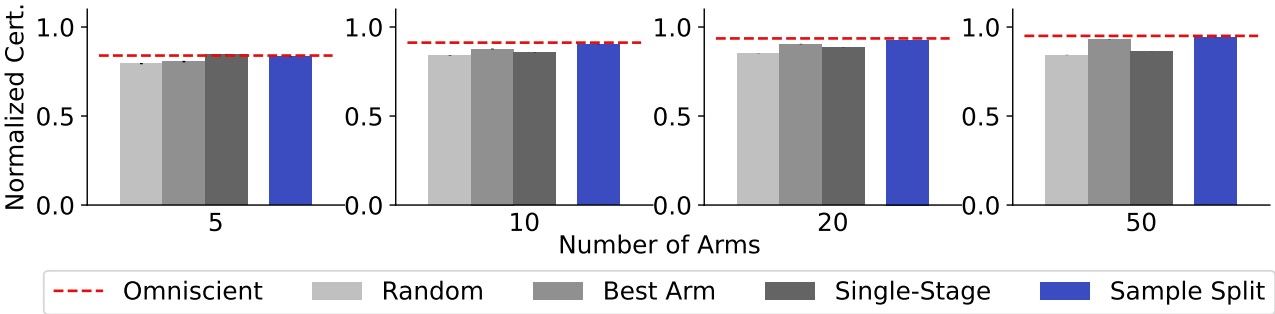

*Figure 11.* We vary the number of arms, $n$, and find that sample splitting algorithms are the best design across choices of $n$. Larger $n$ leads to better performance for best arm designs compared with single-stage designs.

We vary the parameter $n$ in Figure 11 and find that for all values of $n$, the best design is sample splitting. Comparing the other designs, we find that for larger $n$, best arm designs tend to perform better than single-stage designs. This occurs due to the increased importance of pruning bad arms for $n$ large, as more budget is wasted on suboptimal arms if pruning does not occur.

## E. Varying distribution of arm means

We vary the distribution of $\mu$ for various uniform distributions, and plot our results in Figure 12. Adaptive policies perform worse when the average of the uniform distribution is large, which might be because unnecessary time is spent exploring in UCB policies. Moreover, when the distribution of arm means is near one, we find that sample splitting designs can even outperform fully adaptive designs, and are the best-performing designs in that scenario.

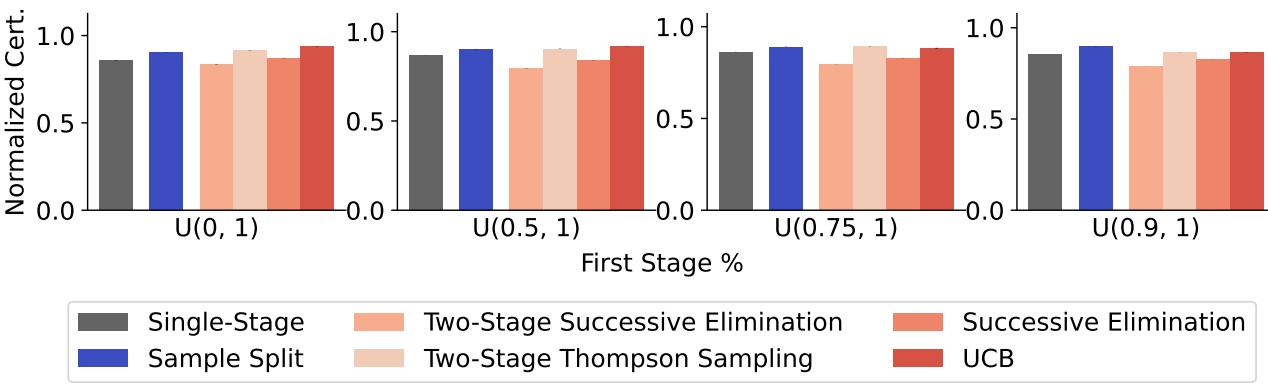

*Figure 12.* We vary the distribution of the underlying arm means distribution, and find similar performance for sample splitting designs across distributions. When the average arm mean is closer to 1, we find that our sample splitting policies can even outperform fully adaptive methods, such as UCB. .

## F. Proofs

**Proposition F.1.** *Let $\Delta_{ij} = \mu_{\sigma_i} - \mu_{\sigma_j}$. Let $\boldsymbol{\mu}$ be such that assumption 3.4 holds. Let $\sigma$ be the permutation of the indexes obtained from sorting the empirical means obtained in the first stage in descending order. Let $c(i) = \sqrt{2log(\frac{2i}{\delta})\frac{i}{s_2}}$. Then, conditioned on $\boldsymbol{X}$ for any top-k policy $\pi^k$ obtain as an output of algorithm 1, the expected value of its certificate $f(\pi^k)$ is bounded below by*

$$f(\pi^*) - \sum_{i=1}^{k^*} \exp\left(\frac{-\left[\Delta_{k^*i} - (c(k^*) - c(i))\right]^2 s_1}{n}\right)$$
$$\times \left[\Delta_{k^*i} - (c(k^*) - c(i))\right]$$
$$- \sum_{i=k^*+1}^{n} \exp\left(\frac{-\Delta_{ik^*}^2 s_1}{n}\right)\left[\Delta_{k^*i} - (c(k^*) - c(i))\right] \qquad (3)$$

*Proof.* As a consequence of Theorem 1 and the stochastic dominance of Bernoulli arms, it suffices to analyze the error against the top-$k^*$ policy. Let $\pi^k$ be the top $k$ policy outputted by algorithm 1. Clearly

$$f(\pi^*) - f(\pi^k) = \mathbb{E}[\max_{i \in \pi^*(X)} \mu_i - c(k^*)] - \mathbb{E}[\max_{j \in \pi^k(X)} \mu_j - c(k)]$$

**Case 1**, $k < k^*$. In this scenario, $c(k^*) - c(k) \geq 0$, hence if $\max_{j \in \pi^k(X)} \mu_j \geq \max_{j \in \pi^*(X)} \mu_j$ that will contradict the optimality of $\pi^*$, furthermore, this is true for every $k < k^*$, thus concluding that $\mu_{\sigma_*(X)} > \mu_{\sigma_i(X)}$ and $\mu_{\sigma_*(X)} = \max_{i \in \pi^*} \mu_j$. Additionally by optimality of $k^*$, $\mu_{\sigma_*(X)} - c(k^*) > \max_{j \in \pi^k(X)} \mu_j - c(k)$. By definition of the alorithm 1, $\overline{V}_{\sigma_k} - c(k) > \overline{V}_{\sigma_{k^*}} - c(k^*)$ then given that $c(k^*) - c(k) < \mu_{\sigma_*(X)} - \mu_k$. In particular:

$$\leq \sum_{i=1}^{k^*} P(\overline{V}_{\sigma_{k^*}} - \overline{V}_{\sigma_i} - c(k^*) + c(i) < 0)[\Delta_{k^*i} - (c(k^*) - c(i))]$$
$$\leq \sum_{i=1}^{k^*} exp(\frac{-2[\Delta_{k^*i} - (c(k^*) - c(i))]^2 s_1}{2n})[\Delta_{k^*i} - (c(k^*) - c(k))]$$

**Case 2** $k^* < k$, $c(k^*) - c(k) < 0$, thus as $\overline{V}_{k^*} - \overline{V}_k < c(k^*) - c(k)$ (Otherwise the index $k$ would not have been picked) then it must be concluded that $\overline{V}_{k^*} - \overline{V}_i < 0$. Then we can bound these terms using the event $P(\overline{V}_{k^*} - \overline{V}_k < 0)$ in which the traditional Hoeffding's bound trick can be plugged.

$$\leq \sum_{i=*+1}^{n} P(\overline{V}_{k^*} - \overline{V}_i < 0)[\Delta_{k^*i} - (c(k^*) - c(i))]$$

$$\leq \sum_{i=k^*+1}^{n} P(\overline{V}_{k^*} - \overline{V}_i - \Delta_{k^*i} < -\Delta_{k^*i})[\Delta_{k^*i} - (c(k^*) - c(i))]$$

$$\leq \sum_{i=*+1}^{n} exp(\frac{-2\Delta_{ik^*}^2 s_1}{2n})[\Delta_{k^*i} - (c(k^*) - c(i))]$$

Note that although $\Delta_{k^*i}$ might be negative, $\Delta_{k^*i} - (c(k^*) - c(i))$ is not as a consequence of the optimality of $k^*$. $\quad\square$

**Lemma F.2.** *Fix an arbitrary policy $\pi$. Define it's top-k counterpart $\pi'$ to be the top-k policy with $k(\mathbf{X}) = |\pi(\mathbf{X})|$. That is, it selects the same number of arms as $\pi$ does for every realization of the trial, but it selects those arms to be the ones with the largest empirical mean. Then $f_{\boldsymbol{\mu}}(\pi') \geq f_{\boldsymbol{\mu}}(\pi)$.*

*Proof.* We conclude that since $\max_{i \in \pi(X)}\{\mu_i\}$ is a non-decreasing function in arguments $\{\mu_i : i \in \pi(X)\}$, first-order stochastic dominance implies (Is equivalent to the expectations matching for every non-decreasing function).

$$\mathbb{E}[\max_{i \in \pi'(X)} \mu_i] \geq \mathbb{E}[\max_{i \in \pi(X)} \mu_i]$$

from which it follows that $f_{\boldsymbol{\mu}}(\pi') \geq f_{\boldsymbol{\mu}}(\pi)$ $\quad\square$

**Theorem 3.7.** *Let $\pi^*$ the policy that maximizes $f_{\boldsymbol{\mu}}$. There exists a top-k policy $\pi$ such that $f_{\boldsymbol{\mu}}(\pi^*) = f_{\boldsymbol{\mu}}(\pi)$.*

*Proof.* Let $\pi^*$ be the optimal policy, as a consequence of Lemma 3.6 then there exists $\pi'$, a top-k policy, such that $f(\pi') \geq f(\pi)$. By optimality the of $\pi^*$ the result follows. $\quad\square$

**Lemma F.3.** *Let $\sigma$ be the descending ordering of arms by empirical mean observed in the first stage. Then, for any $i < j$, $D_{\mu_{\sigma_i(\mathbf{X})}}$ first-order stochastically dominates, $D_{\mu_{\sigma_i(\mathbf{X})}}$*

*Proof.* Fix a given $t$. We compare $\Pr(\mu_{\sigma_i(X)} \geq t)$ and $\Pr(\mu_{\sigma_j(X)} \geq t)$ via a coupling argument where we fix the mean of every position in the permutation except for positions $i$ and $j$. Formally, we decompose

$$\Pr(\mu_{\sigma_i(X)} \geq t) - \Pr(\mu_{\sigma_j(X)} \geq t) =$$
$$\mathbb{E}[1[\mu_{\sigma_i(X)} \geq t] - 1[\mu_{\sigma_j(X)} \geq t]]$$
$$= \mathbb{E}[\mathbb{E}[1[\mu_{\sigma_i(X)} \geq t] - 1[\mu_{\sigma_j(X)} \geq t]|\mu_{\sigma_k(X)}, k \neq i, j]].$$

We will show that the inner expectation $\mathbb{E}[1[\mu_{\sigma_i(X)} \geq t] - 1[\mu_{\sigma_j(X)} \geq t]|\mu_{\sigma_k(X)}, k \neq i, j]$ is always nonnegative. After conditioning on $\mu_{\sigma_k(X)}, k \neq i, j$, there are two possible values for $\mu_{\sigma_i(X)}$ and $\mu_{\sigma_j(X)}$ (i.e., the means of the two arms that did not appear in other positions of the permutation). Denote these values by $\mu_a$ and $\mu_b$ where $\mu_a \geq \mu_b$ refers to the larger of the two. There are two cases. The first case is where either $\mu_a \geq \mu_b \geq t$ or $t \geq \mu_a \geq \mu_b$. Denote this event as $\mathcal{E}_1$. By definition, $\mathbb{E}[1[\mu_{\sigma_i(X)} \geq t] - 1[\mu_{\sigma_j(X)} \geq t]|\mathcal{E}_1, \mu_{\sigma_k(X)}, k \neq i, j] = 0$. The second case is that $\mu_a \geq t > \mu_b$. Denote this event as $\mathcal{E}_2$. Since $i < j$, $\sigma_i(X) = a$ if and only if $\overline{X}_a > \overline{X}_b$, which in turn implies that $\mu_{\sigma_i(X)} \geq t$ if and only if either $\overline{X}_a > \overline{X}_b$ or $\overline{X}_a = \overline{X}_b$ and $r_a > r_b$. Because $\mu_a \geq \mu_b$, the assumption of a stochastic dominance ordering on the arms combined with the independence of the samples implies that $Pr(\overline{X}_a > \overline{X}_b|\mathcal{E}_2, \mu_{\sigma_k(X)}, k \neq i, j) \geq Pr(\overline{X}_b > \overline{X}_a|\mathcal{E}_2, \mu_{\sigma_k(X)}, k \neq i, j)$. Moreoever, whenever $\overline{X}_a = \overline{X}_b$, $\Pr(r_a > r_b) = \frac{1}{2}$ by definition. Since conditioned on $\mathcal{E}_2$, $1[\mu_{\sigma_j(X)} \geq t] = 0$ whenever $1[\mu_{\sigma_i(X)} \geq t] = 1$, we conclude that $\mathbb{E}[1[\mu_{\sigma_i(X)} \geq t] - 1[\mu_{\sigma_j(X)} \geq t]|\mathcal{E}_2, \mu_{\sigma_k(X)}, k \neq i, j] => 0$ and the lemma follows. $\quad\square$

**Theorem 3.8.** *Let $\hat{\pi}$ be the policy obtained by our proposed design using $d$ samples from the posterior $\mathcal{P}(\boldsymbol{\mu}|\mathbf{X})$. Then $f(\hat{\pi}) \geq f(\pi^*)(1 - 1/e - \epsilon)$, where $\epsilon = O\left(\sqrt{\log\left(\frac{1}{\delta}\right)d}\right)$.*

*Proof.* We aim to show that $\hat{\pi}$ is such that $f_{\boldsymbol{\mu}}(\hat{\pi}) \geq f_{\boldsymbol{\mu}}(\pi^*)(1 - 1/e)$. We do so by reducing our problem to an instance of cardinality-constrained submodular optimization, and then note that we can solve such a problem up to a $1 - 1/e$ approximation in polynomial time (Nemhauser et al., 1978).

Let $B^i$ be the set of size $i$ obtained from the algorithm. Let $\hat{Z}_1, \hat{Z}_2, \ldots, \hat{Z}_d$ be $d$ samples of the second stage pulls from the posterior. We use slight abuse of notation, and let $\hat{\mu}(Z_j)$ be the set of empirical means, we then can approximate $f$ as $(\frac{1}{d} \sum_{j=1}^{d} \max_{i \in B} \hat{\mu}(Z_j)) - O(\sqrt{\frac{i}{s_2}})$ with high-probability. Note that $O(\sqrt{\frac{i \log(\frac{1}{\delta})}{s_2}})$ turns our empirical mean into a high probability certificate. We note that our interior optimization problem, $\frac{1}{d} \sum_{j=1}^{d} \max_{i \in B} \hat{\mu}(Z_j)$ is a sum of monotonic submodular functions (the maximum function), and therefore, our entire function is monotonic submodular.

To allow for our whole optimization problem to be monotonic and submodular, we consider all values of $|B^i| \leq n$, and can solve $n$ such submodular optimization problems. We note that the outer function, $-O(\sqrt{\frac{i}{s_2}})$ is constant given $i$. We therefore brute force all $n$ values of $i$, and each such submodular optimization can be solved in $O(nd)$. Our overall problem can therefore be solved in $O(n^2 d)$ to an approximation ratio of $1 - 1/e - \epsilon$ with high probaility, where $\epsilon = O(\sqrt{\log(\frac{1}{\delta})d})$.

$\square$

**Theorem F.4.** *Consider the problem of identifying a high probability certificate using a two-stage sample splitting method. Let there be $n$ arms, with the mean outcome for arm $i$, denoted $\mu_i$, distributed according to a joint prior $\mathcal{P}$. Next, consider the problem of selecting an optimal subset of arms for the second stage, i.e., finding $\pi^*$. Then no polynomial time algorithm can find a policy $\hat{\pi}$, so that $f(\hat{\pi}) > f(\pi^*)(1 - 1/e)$ for all priors $\mathcal{P}$.*

*Proof.* We first sketch a relationship between this problem and the cardinality-constrained instance. Consider the problem of $\max_{|\pi(X)| \leq r} \mathbb{E}[\max_{i \in \pi(X)} l_i]$ for some fixed $r$. By assumption, for fixed $k = |\pi(X)|$, we have that $l_i = X_i - g(k)$ for some function $g$ that does not dependent on $i$. Any polynomial time algorithm that can solve our cardinality-constrained problem can also solve the non-cardinality-constrained problem through an additional factor of $n$ by searching through all values for the budget.

We next demonstrate that no polynomial time algorithm achieves better than a $1 - 1/e$ approximation to the problem of $\max_{|\pi(X)| \leq r} \mathbb{E}[\max_{i \in \pi(X)} X_i]$ when given access to the prior distribution $\mathcal{P}$. We do so by reducing the maximum coverage problem to this.

We consider the maximum coverage problem with sets $C_1, C_2, \ldots, C_n$, where each set has elements taken from a universe $U$ of size $d$. Our goal is to produce a set of sets, $C'$, with at most $r$ sets, that maximizes:

$$\left| \bigcup_{C_i \in C'} S_i \right| \tag{7}$$

Alternatively, our objective can also be viewed as the number of elements in $U$ covered by the sets $C$; that is, we can write the objective as

$$\sum_{i=1}^{d} \mathbb{1}[\exists j, C_j = 1, u_i \in C_j] \tag{8}$$

To perform a reduction to our cardinality-constrained certificate problem, we consider the problem of selecting at most $r$ arms out of $n$ for second-stage evaluation. We let $s_1 = 0$, and only consider a second-stage evaluation. In this situation, the posterior is the prior, $\mathcal{P}$, so we select arms to maximize:

$$\max_{|\pi(X)| \leq r} \mathbb{E}_{\mathcal{P}}[\max_i X_i] \tag{9}$$

The prior distribution, $\mathcal{P}$, denotes the probability of selecting particular outcomes, which correspond to the vector $\mu$. We let the prior in this situation be uniformly distributed according to $d$ vectors, $v_1, v_2, \ldots, v_d, v_i \in [0, 1]^n$. Here, let $d_{i,j} = 1$ if $U_i \in C_j$, where $U_i$ is the ith element in the universe $U$, and $C_j$ is the jth set $C$. Because the expectation is uniform and

taken over $d$ elements, we rewrite the maximization problem as follows:

$$\max_{|\pi(X)|\leq r} \mathbb{E}_{\mathcal{P}}[\max_i X_i] = \max_{\mathbf{b}\in\{0,1\}^n, \|b\|\leq r} \frac{1}{d}\sum_{i=1}^{d}\max_j b_j d_{i,j}$$

Because both $b_j$ and $d_{i,j}$ are 0-1 values, our objective can also be seen as

$$\max_j b_j d_{i,j} = \mathbb{1}[\exists j, b_j = 1 \wedge d_{i,j} = 1] \tag{10}$$

Because our objective is to find the largest such $B$, we drop the constant $\frac{1}{d}$, and rewrite the objective as

$$\max_{\pi(X)} \sum_{i=1}^{d} \mathbb{1}[\exists j, b_j = 1 \wedge d_{i,j} = 1] \tag{11}$$

This objective matches our objective for the coverage problem; if we view $d$ as the matrix denoting which elements $i$ are contained in which sets $j$, then the objective for our certificate problem is the same as the objective for the coverage problem, just off by $\frac{1}{d}$. Therefore, if we can solve the certificate problem for this instance, then we can solve the coverage problem. However, because the coverage problem has a 1-1/e lower bound, our cardinality-constrained certificate problem also has a 1-1/e lower bound (Feige, 1998).

$\square$

