# OpenReview forum: "Data-driven Design of Randomized Control Trials with Guaranteed Treatment Effects"
_ICML.cc/2025/Conference — ICML 2025 poster_

### Official Review · Reviewer_hoJN · 2025-03-04

**Overall Recommendation:** 3

**Summary:**

The paper introduces a novel two-stage design for randomized controlled trials (RCTs) that aims to improve efficiency compared to traditional single-stage designs. In the first stage, all treatment arms are explored uniformly, and a data-driven screening procedure prunes those with low estimated effect. In the second stage, the remaining arms are re-sampled to compute high-probability lower bounds (or certificates) on their treatment effects. The authors provide a theoretical analysis that show under certain conditions a top‑k policy is nearly optimal, and also extend the design to incorporate Bayesian priors.

**Claims And Evidence:**

The paper is clearly written, and the idea is original. The theoretical claims seem justified, at least to the best of my knowledge.

I have reservations about the experimental results. The authors claim that two-stage designs outperform single-stage approaches, yet it is unclear in what sense this improvement is measured. There is limited evidence provided that these designs enhance sample size efficiency or statistical precision—key factors in classical randomized trials, where the width of the asymptotic confidence intervals is the primary concern.

**Essential References Not Discussed:**

To my knowledge, all the essential references are discussed.

**Experimental Designs Or Analyses:**

N/A

**Methods And Evaluation Criteria:**

1. **Evaluation Metrics:**
   The experiments focus exclusively on evaluating the certificate proposed in the paper. In practice, the primary goal in running trials is to reduce the sample size as much as possible—that is, to improve statistical efficiency. It is not clear from the experiments how much sample size saving the method can offer. It would be helpful to see comparisons that directly measure the reduction in sample size needed to achieve a desired level of statistical precision (e.g. confidence interval width).

2. **Single-Stage Baseline:**
   The paper does not explain in detail how the single-stage baseline is implemented, in particular what estimator is used to compute the treatment effects. One might expect that using an Augmented Inverse Probability Weighting (AIPW) estimator, rather than the classic empirical mean, could provide much stronger results. In fact, if one had access to a perfect model for $E[Y|X]$, the AIPW estimator should achieve the same efficiency as the proposed method by effectively imputing outcomes for patients assigned to "bad" arms. Clarification on this point and a comparison with a more advanced baseline would be valuable.

**Other Comments Or Suggestions:**

N/A

**Other Strengths And Weaknesses:**

N/A

**Questions For Authors:**

I have a few questions regarding the experimental evaluation that I hope you could clarify:

1. **Evaluation Metric and Sample Size Savings:**
   - Could you elaborate on why the experimental evaluation focuses only on the certificate as the performance metric?
   - How does the certificate translate into tangible sample size savings or improved statistical efficiency in real-world RCT settings?
   - Can you provide insights or additional experiments that show the efficiency gains from the proposed method as the fixed sample size budget increases?

2. **Details on the Single-Stage Baseline Implementation:**
   - Could you clarify the implementation details of the single-stage baseline used in your experiments?
   - What estimator is used to compute the treatment effects in the single-stage setting?
   - Have you considered or benchmarked the performance of alternative estimators, such as the AIPW estimator?
   - In scenarios where a near-perfect model for $E[Y|X]$ is available, an AIPW estimator achieves the efficiency lower bound by imputing the outcomes assigned to the "bad" arms. How do you expect the performance of a single-stage method using AIPW to compare with your proposed two-stage design in this setting?

**Relation To Broader Scientific Literature:**

It would be valuable to discuss the methods that focus on improving efficiency within the single-stage design e.g. [1,2,3,4,5]. In particular, commenting on whether this line of works can achieve the same efficiency as a 2-stage design in some scenarios. This would give a a more comprehensive view of the trade-offs involved in different trial designs.


[1] Rickard Karlsson, Guanbo Wang, Jesse Krijthe, and Issa Dahabreh. Robust integration of external
control data in randomized trials.

[2]  Lauren Liao, Emilie Højbjerre-Frandsen, Alan Hubbard, and Alejandro Schuler. Prognostic adjustment
with efficient estimators to unbiasedly leverage historical data in randomized trials.

[3] Pierre-Emmanuel Poulet, Maylis Tran, Sophie Tezenas du Montcel, Bruno Dubois, Stanley Durrleman,
and Bruno Jedynak. Prediction-powered inference for clinical trials.

[4] Piersilvio De Bartolomeis, Javier Abad, Guanbo Wang, Konstantin Donhauser, Raymond M. Duch, Fanny Yang, and Issa J. Dahabreh.  Efficient Randomized Experiments Using Foundation Models.

[5] Alejandro Schuler, David Walsh, Diana Hall, Jon Walsh, and Charles Fisher. Increasing the efficiency
of randomized trial estimates via linear adjustment for a prognostic score

**Theoretical Claims:**

I did not check the correctness of any proofs.

---

> ### Author Rebuttal · Authors · 2025-03-31
>
> Dear Reviewer  hoJN
>
> We thank the reviewer for their kind words and enthusiasm for our paper. We address your concerns below.
>
> ### **Claims and evidence/Questions For Authors:**
>
> >I have reservations about the experimental results. The authors claim that two-stage designs outperform single-stage approaches, yet it is unclear in what sense this improvement is measured. There is limited evidence provided that these designs enhance sample size efficiency or statistical precision—key factors in classical
> randomized trials, where the width of the asymptotic confidence intervals is the primary concern.
> >Could you elaborate on why the experimental evaluation focuses only on the certificate as the performance metric?
>
>
> The key point of our paper is that we are pruning arms, not trying to estimate the means of all interventions precisely. Our setting is designed for scenarios where the goal is to obtain a tight confidence interval for the *best* possible policy—not for all arms individually. In fact, our lower bound explicitly characterizes the width of the confidence interval for the best policy only. We intentionally do not aim to produce tight confidence intervals for lower-quality interventions, as they are ultimately discarded by our method.
>
> > It is not clear from the experiments how much sample size saving the method can offer. It would be helpful to see comparisons that directly measure the reduction in sample size needed to achieve a desired level of statistical precision (e.g. confidence interval width).
> >How does the certificate translate into tangible sample size savings or improved statistical efficiency in real-world RCT settings?
>
> Our method should be understood as providing a worst-best case scenario guarantee under a fixed budget. That is, given a fixed budget, we are able to offer a statistically valid guarantee on the performance of a high-impact treatment.
>
> >Can you provide insights or additional experiments that show the efficiency gains from the proposed method as the fixed sample size budget increases?
>
> Figure 2 in the paper illustrates the average value of our certificate as a function of the available budget. As shown in the plot, increasing the total budget reduces the gap between the sample splitting approach and the two-stage policy—demonstrating that sample splitting increasingly approximates the performance of the two-stage design as more data becomes available.
>
> >Could you clarify the implementation details of the single-stage baseline used in your experiments?
> >What estimator is used to compute the treatment effects in the single-stage setting?
>
> The estimate is given by the sample mean of each arm. In the first stage, we use uniform allocation across all $n$ original arms. We then compute the estimator using all collected data—there is no data discarded or ignored in this process.
>
> >Have you considered or benchmarked the performance of alternative estimators, such as the AIPW estimator?
> >In scenarios where a near-perfect model for is available, an AIPW estimator achieves the efficiency lower bound by imputing the outcomes assigned to the "bad" arms. How do you expect the performance of a single-stage method using AIPW to compare with your proposed two-stage design in this setting?
>
> We do not incorporate features in our current design, and therefore the class of methods suggested by the reviewer does not apply in our setting. Nevertheless, we thank the reviewer for raising this point—it would indeed be interesting to explore how such methods compare in an extended version of this work where features/context are available.
>
>
> We thank again the reviewer for their kind words and we look forward to clarify any extra question that arrises.

---

### Official Review · Reviewer_S1mJ · 2025-03-12

**Overall Recommendation:** 2

**Summary:**

The paper studies a setup where one is trying to optimize the design of a randomized controlled trial to identify effective treatments more efficiently. Specifically, they describe a two-stage RCT design/algos to that end.

### update after rebuttal

I thank the authors for their response, which helped me understand some of their results better. I still think UCB could be an interesting baseline in the two-step synthetic setting, where the underperforming treatments based on their upper/lower bounds can be eliminated after the first stage. Further, while I understand authors present empirical results as to the performance with respect to the choice of first stage budget $s$, it is important to have a more nuanced analytical understanding of it in the paper, since it will most likely depend on the inherent variability in the patient outcomes (hence should be chosen in a data driven way itself). For these reasons, I have chosen to maintain my score.

**Claims And Evidence:**

Theoretical claims made in the paper are supported. They are mainly adaptations of existing results from bandit literature and not substantial.

It is not clear what "optimal k" is, and that if authors' algorithm finds this or not. Algorithm 1 is confusing in that the for loop goes from 1 to k, and then k is defined after that for loop, so that probably needs fixing. It seems like $k$ and $s_1$, which is the budget for "first stage" are super critical for the final output/success of the algorithm, but there is no recipe as to how to choose them. For that reason, I do not see how the existence of a top-k algorithm results are interesting/useful (i.e., how do I choose $k$ or $s_1$ is more important).

The sample splitting approach can be a big throw-off for practitioners. RCTs are already small in sample size and throwing some part of the data after the first stage does not seem like the best use of data. Alternative approaches such as cross-fitting could be useful here. Also, it is not clear how this approach would affect subsequent subgroup analysis, which are often the most interesting parts of RCTs. i believe thinking about this carefully and including discussions is critical, as sample splitting will hurt statistical power at that step which already suffers a lot due to small sizes of RCTs.

Methodologically, authors mention that sometimes it is not feasible to wait on the results of an RCT to guide next steps. This I think is another thing that limits the practical adaptation of the algorithm proposed in this paper.

**Essential References Not Discussed:**

There is significant room for improvement for covering the related work. Authors mention that the adaptive trial design literature as an active/big area but they do not really do it justice when it comes to covering it. Off the top of my head I can think of [1-6] below, but there are probably others. The paper's coverage/positioning its contributions is severely lacking in that regard.

[1] Villar, S. S. and Rosenberger, W. F. Covariate-adjusted response-adaptive randomization for multi-arm clinical trials using a modified forward looking gittins index rule. Biometrics, 74(1):49–57, 2018.

[2] Villar, S. S., Bowden, J., and Wason, J. Multi-armed bandit models for the optimal design of clinical trials: Benefits and challenges. Statistical Science, 30(2):199–215, May 2015.

[3] Aziz, M., Kaufmann, E., and Riviere, M.-K. On multi- armed bandit designs for phase I clinical trials. arXiv e-prints, art. arXiv:1903.07082, March 2019.

[4] Atan, O., Zame, W. R., and van der Schaar, M. Sequential patient recruitment and allocation for adaptive clinical tri- als. In Proceedings of The 22nd International Conference on Artificial Intelligence and Statistics, pp. 1891–1900, Apr 2019.

[5] Bornkamp, B., Bretz, F., Dette, H., and Pinheiro, J. C. Response-adaptive dose-finding under model uncertainty. Annals of Applied Statistics, 5:1611–1631, 2011.

[6] Bretz, F., Gallo, P., and Maurer, W. Adaptive designs: The swiss army knife among clinical trial designs? Clinical Trials, 14(5):417–424, 2017.

**Experimental Designs Or Analyses:**

In Figure 4, UCB method seems to work the best by a margin. Can you comment on what is the disadvantage method? It is fairly simple method and complex at all.

Is there a reason why some of the baselines in the real-world experiments (e.g., UCB) are not benchmarked against in the synthetic experiments?

**Methods And Evaluation Criteria:**

see below (exp design... section)

**Other Comments Or Suggestions:**

running title needs fixing
algorithm 1 all lines are numbered 0

**Other Strengths And Weaknesses:**

see above

**Questions For Authors:**

--

**Relation To Broader Scientific Literature:**

The paper uses/adapts existing methods/results to the two stage RCT design problem. I do not see how its contributions would add to broader scientific/ML lit.

**Theoretical Claims:**

I did not check in detail, although the results are a combination of standard bandit results and concentration inequalities that can be found in prob/stat textbooks.

---

> ### Author Rebuttal · Authors · 2025-03-31
>
> Dear Reviewer  S1mJ,
>
> We thank the reviewer for their kind words. We address your concerns below.
>
> ### **Questions/Weaknesses**
>
> >Theoretical claims made in the paper are supported. They are mainly adaptations of existing results from bandit literature and not substantial.
>
> We believe there are several novel theoretical contributions worth highlighting: First, to provide the bandit's approximation guarantee in Theorem 3.2, we developed a new characterization of optimal policies and proved how the problem can be reduced to a search over only top-k policies.  Second, we would like to emphasize that our theoretical results supporting the Bayesian approach are not derived from bandit literature at all. Rather, these contributions are more closely related to submodularity and discrete optimization.
>
> > It seems like $s_1$  and $k$, [...] are super critical for the final output/success of the algorithm, but there is no recipe as to how to choose them. For that reason, I do not see how the existence of a top-k algorithm results are interesting/useful (i.e., how do I choose  or  is more important).
> > It is not clear what "optimal k" is, and if the authors' algorithm finds this or not.
> >[...] RCTs are already small in sample size and throwing some part of the data after the first stage does not seem like the best use of data..
>
> To clarify, we demonstrate that the optimal policy, that is the policy that will yield the highest certificate, is a top-k policy.  Here, the optimal $k$, which we denote as $k^*$, refers to the number of arms that such optimal policy selects for the second round. We agree with the reviewer that the selection of $s_1$ and $k$ is critical to the success of our algorithm. However, there appears to be a misunderstanding regarding the role of $k$: the purpose of our algorithm is precisely to **automatically** select a near-optimal value of $k$, rather than treating it as a user-specified input. To this end, we design our sample splitting algorithm to estimate a value of $k$ with approximation guarantees to the optimal $k$ ($k^*$).
>
> Regarding the choice of the first-stage budget $s_1$: in Figure 3, we analyze how shifting the budget between the first and second stages affects certificate performance. We find that allocating 20–50% of the budget to the first stage yields the best results.
>
> With this clarified, we would like to emphasize that concerns about statistical power in the first stage are somewhat misplaced. The point of our two-stage design is to improve statistical power in the second stage, which directly impacts the final certificate.
>
> > Algorithm 1 [...] the for loop goes from 1 to k, and then k is defined after that for loop, so that probably needs fixing
> We thank the reviewer for pointing out this typo. Instead of k in the for loop it should be $n$ -the original number of arms.
> > It is not clear how this approach would affect subsequent subgroup analysis.
>
> This is a very insightful question. Our view is that by concentrating samples on the more promising arms, we also gain statistical power for conducting subgroup analyses of heterogeneous effects within those treatments. In other words, the approach naturally prioritizes subgroup analysis for the arms with the greatest overall potential.
>
> >Methodologically, authors mention that sometimes it is not feasible to wait on the results of an RCT to guide next steps. This I think is another thing that limits the practical adaptation of the algorithm proposed in this paper.
>
> As we point out in the paper, there are important scenarios—such as drug trials or government interventions with very slow feedback loops—where outcomes can only be observed after several months or even years. In such settings, fully adaptive trials are often infeasible. In summary, while there is indeed a large and growing literature on adaptive trial designs, our work addresses a key practical limitation in their applicability, offering a principled alternative when adaptivity is constrained.
>
> ### **Experimental Designs Or Analyses:**
>
> >In Figure 4, UCB method seems to work the best by a margin. Can you comment on what is the disadvantage method? It is fairly simple method and complex at all.
> >Is there a reason why some of the baselines in the real-world experiments (e.g., UCB) are not benchmarked against in the synthetic experiments?
>
> Our designs are motivated by use cases where delayed outcomes make fully adaptive trials infeasible (e.g., deploying the UCB algorithm in practice). For this reason, UCB is not directly comparable to our setting, which is why it is not included in the benchmarks shown in Figure 1. However, in Figure 4, we do compare our approach to fully adaptive baselines in order to quantify the potential loss relative to an idealized, fully adaptive trial. This comparison highlights the performance tradeoffs while reinforcing the practicality of our method in constrained settings.

---

### Official Review · Reviewer_HuaZ · 2025-03-13

**Overall Recommendation:** 4

**Summary:**

The paper introduces a two-stage randomized controlled trial design to enhance the best possible treatment effect guarantee while reducing wasted resources on sub-optimal arms. In the first stage, a data-driven screening process eliminates low-impact treatments, and in the second stage, the focus shifts to establishing high-probability lower bounds for the best-performing treatment. This approach is simpler than existing adaptive frameworks and can be implemented in scenarios with limited adaptivity. The optimality of top k design is discussed and a practical sample splitting method is developed to determine k. Empirical results demonstrate that this two-stage design outperforms single-stage approaches and is close to the fully adaptive approaches.

**Claims And Evidence:**

Yes

**Essential References Not Discussed:**

No

**Experimental Designs Or Analyses:**

Yes. The numerical experiments are well conducted and analyzed.

**Methods And Evaluation Criteria:**

Yes. I like the methodology part of the paper, which is concise and practical.

**Other Comments Or Suggestions:**

No

**Other Strengths And Weaknesses:**

The idea of the paper itself is not fancy, but I like its practical relevance and theoretical elegance.

**Questions For Authors:**

I have some questions and comments:
1. The use of "treatment effect" in this paper is a bit confusing to me. Typically in RCT, there will be one control arm and many other treatment arms, so the treatment effect should be the gap between any treatment arm and the control arm. In this way, the calculation of the so-called treatment effect certificate should be adjusted --- If I understand the math correctly, the use of concentration inequality can be well extended given that everything is i.i.d; If I miss some key component and the adjustment is actually hard, the authors should change the notion of "treatment effect".

2. In the numerical experiments, it seems that two-stage sample split is worse than two-stage TS, especially the case without prior. I think this observation raises some concerns. It basically says that if we limit the design to be two-stage, actually we should follow TS, though it is tailored for minimizing regret. In addition, I am curious about the performance of fully adaptive TS. The reason is that if fully adaptive TS has the best performance (better than UCB), then there must be some theoretical essence about TS for maximizing certificate.

3. In the Appendix part C, the authors conduct multi-stage designs and obtain the certificate using all-stage data. I am wondering how the certificate is obtained given that the data is not i.i.d.

**Relation To Broader Scientific Literature:**

The work is highly related to best arm identification literature but differs by not choosing ϵ-best arms that exhaustively identify all arms within a fixed distance of the best. Instead, it adaptively selects the number of arms to maximize the final certification strength, which has a substantial value in practice. The design is still based on RCT and the estimation only relies on the last stage, which avoids some critical concern in the adaptive experimental design literature.

**Theoretical Claims:**

I roughly checked the math but not step by step. That said, I feel the theoretical claims are reasonable and valid.

---

> ### Author Rebuttal · Authors · 2025-03-31
>
> Dear Reviewer  HuaZ,
>
> We thank the reviewer for their kind words and enthusiasm for our paper! We address your questions below.
>
> ### **Questions/Weaknesses**
>
> >The use of "treatment effect" in this paper is a bit confusing to me. Typically in RCT, there will be one control arm and many other treatment arms, so the treatment effect should be the gap between any treatment arm and the control arm. In this way, the calculation of the so-called treatment effect certificate should be adjusted --- If I understand the math correctly, the use of concentration inequality can be well extended given that everything is i.i.d; If I miss some key component and the adjustment is actually hard, the authors should change the notion of "treatment effect".
>
> In the Bernoulli setting, the average treatment effect is precisely the mean from the distribution. This is a consequence of the outcome of the model being one (say for the treated) and zero (the untreated). Nevertheless, we do agree, about the need to be more precise in the others settings and we thank the reviewer for the feedback (we will add clarifications in the camera ready version)
>
> >In the numerical experiments, it seems that two-stage sample split is worse than two-stage TS, especially the case without prior. I think this observation raises some concerns. It basically says that if we limit the design to be two-stage, actually we should follow TS, though it is tailored for minimizing regret. In addition, I am curious about the performance of fully adaptive TS. The reason is that if fully adaptive TS has the best performance (better than UCB), then there must be some theoretical essence about TS for maximizing certificates.
>
> Thank you for your insightful observation regarding the Thompson sampling experiments. Our intention with these experiments was to demonstrate that our two-stage approach offers sufficient flexibility to accommodate non-uniform sampling strategies. We appreciate the reviewer noting the effectiveness of Thompson sampling in computing a better certificate. We agree with this assessment and believe this likely occurs in scenarios where the variance in the distributions is substantial relative to the gaps between arms, which can indeed make rapid accumulation of empirical means challenging.
>
>
> >In the Appendix part C, the authors conduct multi-stage designs and obtain the certificate using all-stage data. I am wondering how the certificate is obtained given that the data is not i.i.d.
>
> The final certificate was obtained as in the experiments with two stages with the caveat of only using data from the last stage. The reviewer is absolutely correct that in order to give a theoretical guarantee in this scenario we would need to correct for the dependency inherent of picking a particular subset arms at each stage. However, we believe that we could deliver such a guarantee through only slight modifications which account for the lack of i.i.d-ness, and that the final results would not differ significantly.
>
>
>
> ### **Other Comments Or Suggestions**
>
> We thank the reviewer for the thorough read of our paper and identifying formatting mistakes, and will update our manuscript accordingly.

---

### Official Review · Reviewer_Kupv · 2025-03-14

**Overall Recommendation:** 3

**Summary:**

This study proposes a two-stage RCT design aimed at improving efficiency in treatment effect estimation by reducing unnecessary resource allocation to sub-optimal treatments. The idea is pretty straightforward: use top-K policy to screen the inferior arms, then put more resources on the better arms. The authors derive optimal designs, demonstrate their feasibility through sample splitting, and provide empirical evidence of improved performance over single-stage approaches.

**Claims And Evidence:**

Yes, the claims made in the submission are supported by clear and convincing evidence.

**Essential References Not Discussed:**

Not aware of any.

**Experimental Designs Or Analyses:**

Yes. I checked all the experiments. Some questions are raised in weakness discussions.

**Methods And Evaluation Criteria:**

Yes.

**Other Comments Or Suggestions:**

1. line 63, right column: the references follow an unnecessary punctuation
2. line 197, left column: what is the "=0" about?
3. line 166, right column: this line is a bit messed up

**Other Strengths And Weaknesses:**

Strength: it is nice to see the top-K policy being applied to the two-stage experiment. Also interesting to notice the application with the Bayesian thinking by incorporating priors.

Weakness:
1. In Alg 1, there is data splitting procedure. How much is this impacting the efficiency of the algorithm? Is it possible to do a cross fit type of algorithm? Sampling splitting is typically an inferior choice as data points in many RCTs are pretty expensive.
2. in practice, researchers are also making decisions about choosing $s_1$ and $s_2$. The author had some discussion on the tension between these two parameters, but what is the practical guidance from the theory for the best choice of $s_1$ and $s_2$?
3. For incorporating the prior, there is always a $1-1/e$ gap guaranteed by the lower bound. Is that saying (theoretically) incorporating prior is a slightly worse strategy compared with sample split? This is counter-intuitive as I would assume prior information on the parameters can provide more guidance for policy designing.
4. For simulation (Compare against Adaptive designs), Sample Split is actually doing close to (a little bit worse than) Two-stage Thompson sampling (TS). Is this a general phenomenon or simply due to the experimentation design?

**Questions For Authors:**

Please see the weakness part.

**Relation To Broader Scientific Literature:**

This work is a fair addition to the literature of two-stage adaptive experiments.

**Theoretical Claims:**

I have checked the proof of the main theorems (3.7 and 3.8), which look correct to me.

---

> ### Author Rebuttal · Authors · 2025-03-31
>
> Dear Reviewer  Kupv,
>
> We thank the reviewer for their kind words and enthusiasm for our paper. We address your concerns below.
>
> ### **Questions/Weaknesses**
>
> > In Alg 1, there is data splitting procedure. How much is this impacting the efficiency of the algorithm? Is it possible to do a cross fit type of algorithm? Sampling splitting is typically an inferior choice as data points in many RCTs are pretty expensive.
>
> We thank the reviewer for their valuable suggestion. We focus on the sample split algorithm for ease of presentation but it can be easily extended to a cross-fit style of approach. We run a small experiment with this, comparing the results of cross validation to that of sample split, using the original settings from Figure 1 (with equal budget between stage 1 and stage 2). We find that sample splitting performs as well, if not better, than cross validation. Using cross validation with k=2 performs slightly worse (~1% worse), while for k=3,4,5, we find that it performs ~4% worse compared to sample splitting. This occurs because cross validation sacrifices test data points for training ones; for example, doing k=4 folds results in ¼ of the data being used for evaluation, and ¾ for some analog of training. As a result, we find that the size of these evaluation data points are critical to reducing variability.
>
> > In practice, researchers are also making decisions about choosing  $s_1$ and $s_2$. The author had some discussion on the tension between these two parameters, but what is the practical guidance from the theory for the best choice of   $s_1$ and $s_2$ ?
>
> The reviewer raises a very important inquiry. In Figure 3 of the paper, we compare the impact of shifting budget between $s_1$ and $s_2$ on the performance of various policies for certificate generation. We find that having the first stage occupy between 20-50% of the budget allows for optimal performance. This occurs because the first stage needs to have a large enough size to eliminate sub-optimal arms, while not being too large that the second stage is small. Intuitively, the first stage should be just large enough so that any suboptimal arms can be eliminated.
>
> > For incorporating the prior, there is always a $1 - 1/e$ gap guaranteed by the lower bound. Is that saying (theoretically) incorporating prior is a slightly worse strategy compared with sample split? This is counter-intuitive as I would assume prior information on the parameters can provide more guidance for policy designing.
>
> We note that the $1-1/e$ bound is not against the absence of a prior, but rather the optimal selection of arms even with the prior.  In essence, while our bound is worse, our comparison point is against a better opt value. Naturally the reviewer is right to point out that a better prior will lead to better finite sample guarantees for the Monte Carlo approximation and thus probably a less pessimistic lower bound.
>
> > For simulation (Compare against Adaptive designs), Sample Split is actually doing close to (a little bit worse than) Two-stage Thompson sampling (TS). Is this a general phenomenon or simply due to the experimentation design?
>
> It is likely that for our particular scenario -- where the posterior properly captures the variance of the arms -- Thompson sampling is better than uniform sampling. However, this might not be the case if the gaps between arms are big enough. Such a scenario will lead to a faster accumulation of the empirical means than the convergence of Thompson sampling.
>
> ### **Other Comments Or Suggestions**
>
> We thank the reviewer for the thorough read of the paper and pointing out this formatting mistakes. We plan to correct these in our final version of the paper.

---

> > ### Comment · Reviewer_Kupv · 2025-04-03
> >
> > Thanks for the careful responses.
> >
> > 1. To note, when I was saying cross-fitting, I was talking about: splitting data into two-halves U and V, then use U as training and V as validation, then switching the role of the sets by using U as validation and V as training, then combine the results. I wasn't sure if you are referring to such as practice when mentioning "cross-validation". Cross-fitting is pretty common in analyzing property of estimators while keeping full use of the data.
> >
> > 2. The suggestion for choosing $s_1$ and $s_2$ sounds good to me.
> >
> > 3. The clarification that we are comparing against a better optimal makes sense to me.
> >
> > 4. The explanation makes sense to me. I am happy to see maybe a more rigorous (theoretical) analysis/comparison of these methods in some future work.

---

> > > ### Author Response · Authors · 2025-04-07
> > >
> > > Thank you for your response. We find your suggestions very useful, and we will incorporate them into our final manuscript.
> > >
> > > In response to your mention of cross-fitting, we agree with your definition and have experimented with that method, which we discussed in the earlier reply (under the name of "cross validation"). Let us know if there are any other questions we can answer.

---

### Decision · Program_Chairs · 2025-05-01

**Decision:**

Accept (poster)

**Comment:**

The reviewers largely agreed that the authors presented an elegant solution to an important problem and that the work presented here can help improve efficiency in treatment effect estimation by reducing unnecessary resource allocation to sub-optimal treatment